# Urban Consumer Trust and Food Certifications in China

**DOI:** 10.3390/foods9091153

**Published:** 2020-08-21

**Authors:** Roberta Moruzzo, Francesco Riccioli, Fabio Boncinelli, Zhaozhong Zhang, Jinjin Zhao, Yaojia Tang, Lara Tinacci, Tommaso Massai, Alessandra Guidi

**Affiliations:** 1Department of Veterinary Science, University of Pisa, 56124 Pisa, Italy; roberta.moruzzo@unipi.it (R.M.); lara.tinacci@unipi.it (L.T.); tommaso.massai@unipi.it (T.M.); alessandra.guidi@unipi.it (A.G.); 2Department of Agriculture, Food, Environment and Forestry, University of Florence, 50144 Florence, Italy; fabio.boncinelli@unifi.it; 3Research Institute of Regulation and Public Policy, University of Finance and Economics, Hangzhou 310018, China; magiczhang625@163.com (Z.Z.); jinjinzhao@zufe.edu.cn (J.Z.); yaojiatang@zufe.edu.cn (Y.T.)

**Keywords:** logit model, consumer’s trust, food safety, food certification policy, food legislation, consumers’ protection

## Abstract

China has experienced frequent food safety incidents that have undermined consumer trust in the food supply chain. To overcome this problem, China requalified the legislative framework and adopted a comprehensive food certification system over the years. Here, we investigated the influences of food traceability and Chinese certifications (QS/SC—food quality safety market access/production system, hazard-free, green, and organic) on Chinese consumer trust of food safety for different types of products: fish, meat, milk, eggs, and rice. Data were collected through face-to-face surveys conducted in rural and urban Chinese areas. With a sample of 757 questionnaires, we ran a logit model. The results show consumers’ uncertainty and skepticism of certifications guaranteeing food safety attributes, especially for animal-based products. We found that price is used as a cue of safety by Chinese consumers. Individuals with higher education seem less influenced by certifications and other cues included in the analysis. The findings demonstrate that Chinese policy makers should implement new strategies to enhance consumer food safety trust, and design policies by considering different categories (e.g., vegetables, meat, fish, etc.) of food.

## 1. Introduction

The fast development of the Chinese food industry has been accompanied by frequent food safety incidents [1,2] that have undermined consumer trust in food safety [3,4,5,6] and have limited the international reputation of the Chinese food industry [7,8,9,10,11,12]. Many studies reported weaknesses in China’s food safety system, which are mainly related to poor law enforcement and inefficient supervision of governmental agencies [13,14,15]. To enhance consumer trust in food, in 2003, the Chinese government introduced the food quality safety market access system (QS system), meaning “authorized manufacturing for enterprises”, which is the industrial product manufacturing license. Only after applying for this license can food enterprises be allowed to enter the market for processing or trading [16]. In 2018, the QS system was replaced by the Sheng Chan (SC, which means “production”) system [17], which was released by the government. These licenses consist of the letters SC followed by 14-digit numbers providing information about the product and the producers. To indicate QS/SC licensing, we only use “SC” in this paper. This change in policy emphasized the responsibilities of enterprises in ensuring food safety instead of government regulation.

As declared in 2015 by China’s State Council, food safety is a top priority; the government has tried to improve the legislative framework and its implementation. In 2015, the food safety law was rewritten, revised in 2018, and published in October 2019 as the Regulation on the Implementation of the Food Safety Law [18,19,20]. Along with the law requirements, to enhance and ensure consumer trust in food quality and safety, the Chinese government adopted a comprehensive food certification system [21], principally organized by product certifications. Product certification should provide a signal to consumers that the food with this label incorporates certain features, thus eliminating the risk that consumers face when purchasing food products with features that consumers cannot verify.

As argued by Ni [22], product certifications in China mainly include the hazard-free food certification system, green food certification system, and organic food certification system, which have different approval rates and trust among consumers [6,23,24,25]. Jin et al. [26] reported that certification of final products alone cannot assure food safety regarding consumers’ perception of food safety.

Liu et al. [27] defined hazard-free food and green food certification. The first is related to food safety, where the quantities of products dangerous to human health, such as heavy metals, pesticides, and fertilizers, are controlled by national laws. Green certification refers to food obtained using limited and controlled quantities of livestock and poultry feed additive, chemical pesticides, and fertilizers. The green food label refers to a category of food that is grown in a safe and ecologically sound manner. There are two standards for green food: the “A” grade green, which represents a transitional level between conventional and organic food, allowing restricted use of chemical fertilizers and pesticides, and the “AA” grade green food, which represents full organic status.

Organic food is “certified to international standards such as the International Federation of Organic Agriculture Movements (IFOAM) and hence genetically modified ingredients are shunned on ideological grounds” [28]. Descriptions of the three kinds of certifications together with their comparisons are provided in Table 1.

These certifications have been widely used in Chinese academic studies as representative of safe food [31,32,33], intended as the absence of harmful or undeclared biological, chemical, or physical contaminants; they have their respective labels printed on the food packages for identification, together with the certification authority [34]. Notably, despite these certifications (hazard-free, green, and organic) being mainly related to the quality of food, they are strongly implicated with the safety, and many consumers identify them with the safety concept. As argued by Riccioli et al. [35], users identify safer food with high-quality food, and the two concepts are often highly correlated. In particular, Grunert [36] studied this correlation and identified the complexities involved, considering how consumers make their own judgments about the safety and quality of a product. Nicolas et al. [37] studied this correlation, stating that a higher level of food safety is directly related to quality improvement.

Here, we investigated the influences of product certifications on Chinese consumers’ choices of safe food products, analyzing their impacts on different types of products. The preference of Chinese consumers for other food-related attributes, such as traceability information or price, as well as their preferred place of sale (traditional supermarkets vs. grocery stores) for purchasing safe food, were also studied to compare these attributes with certifications.

This article is organized as follows: Section 2 describes the current state of Chinese consumers’ knowledge of food certification, presenting their different attitudes about the quality of safe food and the certification labels. Section 3 describes the materials and methods, Section 4 outlines the results, and Section 5 provides the discussion and conclusions.

## 2. Chinese Consumers’ Knowledge and Safety Perception about Food Certification

Consumer knowledge of safe food concept is still quite low in China [27]. Indeed, they generally have positive attitudes toward safe food but a limited ability to recognize it and identify it as the hygienic quality of a product. Nearly one in two Chinese consumers have little or no confidence in the food certification system [34]. Past studies [38,39] suggested that international certified labels enjoy more trust from the Chinese consumers than their local competitors. Consumers’ preferences and emphasis on different attributes linked to certification system vary depending on location or country [7] and are influenced by the degree of their trust in the government’s supervision of food safety and food labels [40].

Despite the importance of the Chinese food industry and policymakers understanding Chinese consumers’ awareness of food safety and their preferences for food safety information attributes, few studies focused on Chinese consumers’ preferences and behavior with regard to food safety attributes [9,41].

The literature shows that Chinese consumers use the safe food logos on products to identify safe food [27]. Despite the level of knowledge of safe food logos being still too low, adding the certification mark to the food label is becoming an important method to prove food quality [42]. The report produced by the Forum on Health, Environment, and Development states that most surveys showed that Chinese consumers are willing to pay a premium for product attributes indicating quality, principally some form of labeling or certification [43].

Zhang and Wu [44] reported that 35% of consumers can recognize the more common green food labels in the market, less than 10% of consumers were able to identify the organic food label, and only 0.4% of consumers were able to recognize all the safe food labels. Jin and Zhao [45] reported that less than half of consumers in Zhejiang province used the green food label to identify green food, 21% identified green foods by personal feeling, and 18% by food brand. Zhang [46] showed that a large majority of consumers depended on advertisements, vendor guarantees, and friends’ recommendations to identify green food; only 13% used the green food label. Zhang and Wu [44] found that only 40% of consumers in Chengdu (Sichuan province) were able to recognize the green food label, and 64% of consumers did not know that two levels (A and AA) of green food existed. One exception to this situation was reported by Liu et al. [47], who found that a majority (74%) of consumers who were aware of organic pork were able to recognize its label and knew it was officially certified.

The literature also describes different attitudes (sometimes positive and sometimes negative) among Chinese consumers about the quality of safe food and the certification labels. Zhang and Wang [48] reported that in their in-depth interviews, all participants doubted whether green food companies strictly control the production area, process, transportation, and storage. Wang et al. [49] reported that 71% of consumers were dissuaded from buying green pork by its high price (48%), quality and safety issues (22%), lack of availability (17%), and a lack of confidence in the certification label (14%). Wang et al. [50] found that consumers had a low willingness to pay (WTP) for safe food. The reason is that the standard for obtaining safe food certification is relatively low, giving consumers an impression that pork with a safe food logo is minimally different from pork with no certification. Additionally, Liu et al. [27] reported that Chinese consumers have difficulty in identifying safe food certificates and labels.

Green and organic food labels are associated with reports of fake organic products, false labels, and bribing of inspection officials [51,52]. Simultaneously, few products are covered by safe food labels [53].

Other authors found a different attitude of consumers toward certifications. Bekele [30] reported that consumers trust certified green food labels. Green food certification serves as a reassurance to both domestic consumers and to international food manufacturers sourcing ingredients in China. For other authors, the primary driver of demand for green food is the lack of confidence in the safety of Chinese produce [33].

In a large-scale study of the adoption of organic food by Chinese consumers, healthiness, taste, and environmental friendliness were found to be important attributes. Health concerns outweigh environmental concerns in terms of influencing the purchase intentions of Chinese consumers [54]. Consumers with a higher level of concern about vegetable safety pay more attention to the nutritional health and pollution hazards of foods (hazard-free) when they purchase food [55].

Ortega et al. [56] found that concerning Ultra High Temperature (UHT) milk, consumers most value government certification followed by a national brand. The recent milk-safety incidents that affected various nationally recognized brands in China have sparked consumer interest in government certification efforts, especially as they pertain to the monitoring and supervision of branded firms and products.

## 3. Materials and Methods

### 3.1. Questionnaire Design and Survey

The questionnaire was administered to a sample of Chinese consumers in Zhejiang province. Nine of the 11 cities (and surrounding areas) in the province were selected: Hangzhou, Ningbo, Shaoxing, Jinhua, Quzhou, Taizhou, Jiaxing, Huzhou, and Lishui. Two cities were excluded: Wenzhou was excluded because of the high proportion of immigrants, and Zhoushan was excluded because it is built on several islands, which would have complicated the data collection, as fish is as the food mainly consumed in the city.

The questionnaires were administered in January 2017 by specially trained students through random sampling of residents. A total of 1000 questionnaires were collected, of which 757 (75.7%) were used in the analysis; 243 (24.3%) questionnaires were excluded due to incomplete answers. Considering recent similar studies [57,58,59], this size was adequate for our analysis.

To achieve the maximum possible variation, the survey was administered on different days and at different times and places. The interviewers approached the respondents in 3–5 residential districts in the main urban areas and 3–5 villages of the selected cities. According to the proportion of urban and rural population in Zhejiang province, the sample included 30% rural households and 70% urban households. Participants were voluntary, and they did not receive any kind of compensation or gift. Participants when started their interviews were not informed about the final aim and the topic of research.

The questionnaire (originally written in English and then translated into Chinese), created in collaboration with faculty members of the Zhejiang University of Finance and Economics, was divided into the following phases for a total of 36 questions:The first part requested the socio-demographic information of the respondents;The second part asked questions regarding eating habits (e.g., the frequency of consumption of different types of product or the different reasons influencing their purchase);The third part asked about the safety perception of different categories of food such as rice, milk and milk products, eggs, meat, fish, vegetables, and fruit. Questions regarding certifications were asked in this section as well;The fourth part queried regarding the quality of food (same categories as above).

In the last two sections, questions regarding the willingness to pay a higher price for safer or higher quality staple food were asked. As WTP is the maximum amount an individual is willing to pay to receive an improvement or to avoid a loss in their level of well-being [60,61,62,63], through this investigation, it was possible to analyze how the price of a product is related to its safety (price safety cue).

Specifically, concerning the variables used in the analysis we asked the following questions: (i) How much confidence do you have in the safety of the following food categories (staple food, milk and derivates, eggs, meat, fish)? (ii) What is the meaning of this mark (SC) for you? (iii) Which of the following certifications (green, organic, and hazard free?) do you know? (iv) Where do you usually buy food? (v) How often do you read the list of ingredients of the products that you buy? (vi) how much do you think that the price of a product is related to its safety? For more details, see the Appendix A.

In this analysis, we only used a part of the data gathered with the survey, as some were already published in [35].

### 3.2. Econometric Model

In our analysis, we modelled the probability of perceiving a certain category of food as safe according to eight covariates: if the respondents know green, organic, and hazard-free certifications and if the respondents associate foods with the SC logo as safe. As control variables, we included the place where respondents usually complete their food purchasing, if respondents usually read the ingredients of foods, and if they perceive price as a cue of food safety. We ran our model for five food categories: fish, meat, milk, eggs, and rice. The first four categories are animal-based food and represent risky food in terms of contamination and disease contagion. We also included rice since it is the basic staple food for the large part of Chinese and Asian consumers. The probably of perceiving the *j*th food category as safe is given by the logit of the multiple logistic regression model:*P_i_*(*i*th consumer confidence in safety of the *j*th food = 1|X) = *e^g^*^(x)^/1 + *e^g^*^(x)^(1)
where *j* is fish, meat, milk, eggs, or rice, and:*g*(x) = *β*_0_ + *β*_1_*SC* + *β*_2_*green* + *β*_3_*organic* + *β*_4_*hazardfree* + *β*_5_*supermarket* + *β*_6_*ingredient* + *β*_7_*traceability* + *β*_8_*pricecue* + *ε_i_*(2)

All the variables in Equation (2) are dummy variables; *SC* was equal to 1 if the respondent perceived food with *SC* as safe; *green, organic,* and *hazardfree* were equal to 1 if the consumer knew the green, organic, and hazard-free certification, respectively, and was 0 otherwise; *supermarket, ingredient, traceability*, and *pricecue* were equal to 1 if the consumer usually purchases food in supermarkets, usually reads food ingredients, knows the certification for traceability, and perceives higher price as a cue of food safety, respectively; and *ε* is a random error term.

We included the variable *ingredient* to control for respondents who pay more attention to food and so may behave differently compared with other consumers. For the same reason, we included the place where respondents usually purchase their usual food as a control variable. Supermarkets and supermarket chains should have well-established procedures and controls to ensure food safety. Therefore, we hypothesized that the supermarket consumers would have less need for certifications.

We repeated the models in Equation (2) including interaction variables between the covariates and two sociodemographic variables, age and level of education, to detect consumer heterogeneity according to respondent features. Age and level of education were also dummy variables. Age was equal to 1 if the respondents were under 30 years old and 0 otherwise. Finally, education was equal to 1 if the respondent had at least graduated and 0 otherwise. The hypothesis was that more educated people have higher cognitive ability and thus are more likely to use formal certification system, as are younger individuals.

## 4. Results

The sample was slightly oriented toward younger and more educated individuals, with an acceptable variation between the panels (groups) in the sample. Table 2 reports demographic statistics: 44% of interviewees were aged between 20 and 29 years, almost 60% were female, and 65% had graduate degrees. The largest share of respondents, more than 60%, usually purchased food in the supermarket, and 37% usually bought food at local markets and small retailers.

Almost two-thirds of respondents reported they usually read the ingredients but, in general, few knew the certifications. Less than half of the sample knew the organic certification, and only 12.17% knew traceability. Table 3 shows the estimates from the five logit models for the selected foods.

Hazard-free certification and traceability never influenced the probability that consumers perceive the selected foods as safe. None of the rest of the certifications included in the models affected this probability for all selected products. Therefore, when a certification plays a role in influencing consumer perception of food safety, it is product specific.

Two factors seemed more effective as a safety cue: SC and price. The first one was significant for all the five models but with a lower degree of statistical confidence for fish and milk. The latter is the tool that had the greatest impact on food safety because it was the predictor with the largest coefficient, and it was always statistically significant. Finally, control variables, traceability, place of usual shopping, and reading the ingredients never impacted food safety perception.

Looking at the results for each product category, we found that for fish and milk none of the examined certifications had statistically significant coefficients with a confidence greater than 90%. Estimates for meat and eggs revealed that the presence of the organic certification was synonymous with safety for respondents, despite the coefficient being statistically significant only at 90% for eggs.

The pseudo *R*^2^ values were all very low, ranging from 0.02 to 0.03 for fish, meat, milk, and eggs. The pseudo *R^2^* for rice was almost double, with a value of 0.07. This result confirmed that certifications play a small role in determining food safety trust amongst Chinese consumers, especially for animal-based products.

To detect if the role of certifications was heterogenous between different groups of the population, we ran logit models including a set of interaction terms between the covariates and age and education level. The results are shown in Table 4.

Organic certification had a statistically significantly influence with a positive sign for the safety perception for fish, rice, and eggs. The safety perception of meat and milk was never influenced by any of the selected certifications. This model specification confirmed that for all food categories, the price strongly influenced the safety perceived by respondents. However, the interaction terms with age and education were statistically significant with a negative sign. This implied that educated people as well as younger individuals are less oriented toward use price as a safety cue.

Considering the individual food categories, organic certification produced a low perception of buying safe fish and eggs. For those with a high educational level, the organic certification had an opposite effect as the interaction coefficient was negative and statistically significant. Therefore, educated individuals perceive organic fish and eggs as less safe than conventional products. Meat and milk were not influenced by any of the certifications. We found no statistically significant interactions for milk and meat. This indicated that no certifications had a positive impact on consumer safety perception for these products, and this conclusion did not vary according to age or educational level.

The model specifications in Table 4 confirm that price was the main determinant affecting consumer perception of buying safe food. However, individuals with higher education were less willing to perceive expensive food as safer.

In general, education was a more discriminant factor than age. The interactions with this last variable were almost not statistically significant, except for the green certification for eggs, which had a stronger impact on younger individuals.

The food where the certification logos was most related to the perception of safety was rice, whereas certification logos on fish and milk were least related to the perception of safety. After the interaction variables of age and level of education were introduced, almost all the certification variables were not significantly positive related to the dependent variable.

## 5. Discussion and Conclusions

In this study, we investigated the determinants of food safety perception of Chinese consumers. In particular, we examined the role of food certifications to better understand consumer awareness of their existence and consumer preferences for food safety information attributes. The limited literature [29,64] and its controversial results [12] reflect the uncertainty in China on this issue, but this information is central for policymakers and the food industry. Our research confirmed that certifications play no clear role in the food safety choices in consumers at large. The price persists as the most important driver of food safety choices as mentioned previously [35,65]; however, our results showed a clear interest toward certified organic food [66], especially in staple foods, like rice, or frequently used foods, like eggs.

In general, however, we found heterogeneous consumer reactions, focusing more on products rather than certifications; in particular, we found a greater mistrust of food of animal origin as confirmed by the low or lack of interest in certifications in meat, fish, and milk, as already found by other authors [12,67].

This finding led us to think of a certain skepticism rather than disinterest in safety attributes and, consequently, that there is a strong need for the government to review the food certification policy and to identify specific solutions for animal-origin food, which are more often the source of major food scandals.

We found that the food with certification logos most related to the perception of safety was rice. This could be due to rice being the staple food in China; therefore, consumers have considerable familiarity with this kind of food. As mentioned previously [68], plant products in China are consumed in larger quantities and at higher frequencies, among of which rice is likely to be the most typical plant product that faces serious food quality and safety issues. The foods where the certification logos were least related to the perception of safety were fish and milk. It is not surprising that the consumers are skeptical about the safety of dairy products after the past dairy safety crisis (see, among others, [1]).

In general, SC certification seems to be the most statistically significant, even if, in some cases, with a low level of confidence, confirming that the government should be the central actor in this requalification of the certification systems, which should also include control of their correct application and transparent communication to consumers.

However, some of our results appear contradictory, for example, by comparing the response to the SC certification with respect to the certified product. Although consumers considered the SC brand safer and more trustworthy, this trust decreased when directed toward the product, preferring other certifications. This can be explained by the fact that in a country with frequent scandals and accidents, it is normal for the consumer to identify the government as the authority that guarantees their safety against producers [69].

Specific concerns arise from multiple factors. Rice’s example is particularly explanatory. Zhu et al. [70] explained that consumer anxiety about rice, and the grain system in general, is connected to different causes: poor knowledge, inability to identify grain quality, high incidence of food-related chronic diseases and cancer, environmental pollution, the minimal efficiency of governmental control, and social media amplifying and distorting news about food safety and food scandals. In this scenario, it is understandable how the consumer would look for further reassurances, such as provided by green and organic certification, in accordance with our results about rice and the other products examined in our study. These results suggest once again that the government should assume a central and well-defined role to construct a clear and transparent certification system that is reliable for the consumer.

Food incidents and widespread information asymmetry discourage consumers from purchasing certified products, especially for those involved in large food scandals. Trust is the most important factor that hinders the functioning of the food certification system [34], and the general lack of trust toward the government and food industry contribute to the failure in attaching importance to voluntary certifications.

Notably, this study was addressed to a wide audience such as citizens, academic researchers, and policymakers, as we explored a transversal theme of common interest. In particular, the lack of strong relationship between brands and animal-based products may be of interest both to researchers (e.g., with the aim to consider different research models according to the different types (animal-based vs. plant-based) of products) and citizens, as it is an increasingly important and timely topic. At the same time, the findings offer policy makers guidance for orienting their efforts toward the activation of specific brands. Our findings can be therefore useful for further research and to guide the government and industry in the actions they have to undertake. According to our results, to associate food safety perception with certifications, institutions need to regain trust from citizens, and the credibility and transparency of food certification bodies need to be increased. Simultaneously, the food industry needs to increase its responsibility and open a proactive dialogue with the government and consumers. The 2019 Regulation on the Implementation of the Food Safety Law demonstrates the government’s awareness of these needs and offers a chance to requalify the sector. The new regulation emphasizes and strengthens the main responsibility of the enterprises and elevates the deliberate release of false information or the concealment of the truth about food products to the grade of severe breach. The new regulation gives the government the responsibility to integrate food safety knowledge into national education and establish a food safety risk exchange mechanism to clarify the contents, procedures, and requirements of information exchange on food safety to raise food safety awareness in society and avoid misleading concerns.

In conclusion, in the context we described, Chinese consumers demand safe food but do not completely trust food certifications as a guarantee of safety. This mistrust could be overcome by the current legislation that has created the premises for a responsible, fair, and transparent food market environment; regulatory enforcement and its implementation remain the challenges.

## Figures and Tables

**Table 1 foods-09-01153-t001:** Comparison of hazard-free food, green food, and organic food certifications.

Certification	Hazard-Free Food Certification	Green Food Certification	Organic Food Certification
Logo	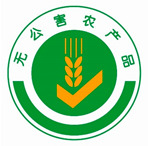	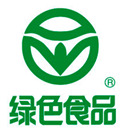	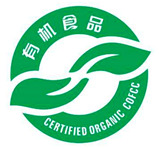
Definition	No pollution, no toxicity, safety, high quality food, clean environment, production technology compliance, control of harmful substances content, use of non-pollution food after examination and approval	Agricultural products grown, produced, or processed in a non-polluting environment, contents of toxic and harmful substances strictly controlled, and conform to the national safety food standards.	No chemical synthetic pesticides, veterinary drugs, feed additives, etc.; no genetic engineering techniques; and specific marks shall not be used for production in accordance with prescribed technology.
Strictness	Low (referring to general domestic food safety standards)	Strict: A-level; Stricter: AA-level (referring to Food and Agriculture Organization of the United Nations (FAO) and World Health Organization—(WHO) standards)	Strictest (referring to European Union and IFOAM standards)
Laws and regulations	Measures for the Administration of hazard-free food products, agricultural product quality and safety law	Green Food labelling Act	Administrative Measures on Organic Product Certification
Label	Printed on the food package for identification	Printed on the food package for identification	Printed on the food package for identification
Validity of certification	3 years	3 years	1 year
Operation year	2001	1990	1994 (Since 2004, there has been unified organic product standards in China)
Type of certification	Certification of land, practices, and products	Certification of land, practices, and products	Certification of land, practices, and products
Certificate authority	The Center for Agri-food Quality and Safety, Ministry of Agriculture of PRC	China Green Food Development Center, Ministry of Agriculture of PRC	Certification and Accreditation Administration of PRC

Source: Own elaboration from [27,29,30].

**Table 2 foods-09-01153-t002:** Descriptive statistics of the sample.

Variable	Description	Relative Frequency (%)
Age (years)	16–19	15.06
	20–29	44.25
	30–39	12.55
	40–49	18.49
	50–60	5.28
	>60	4.37
Sex	Male	40.29
	Female	59.71
Education Level	High school	29.73
	Graduate	64.99
	Postgraduate	5.28
Consumer reads ingredients	Yes	63.01
	No	36.99
Consumer trusts SC logo	Yes	68.82
	No	31.18
Consumer knows hazard-free certification	Yes	37.25
	No	62.75
Consumer knows green certification	Yes	66.58
	No	33.42
Consumer knows organic certification	Yes	43.73
	No	56.27
Traceability	Yes	12.17
	No	87.83
Price safety cue	Yes	44.06
	No	55.94
Purchase place	Supermarket	63.01
	Local market	28.01
	Small retailers	8.98

**Table 3 foods-09-01153-t003:** Influence of brands on the food categories examined.

Variables	Coefficients				
	Fish	Meat	Milk	Eggs	Rice
SC logo	−0.34 *	−0.38 **	−0.33 *	−0.37 **	−0.42 **
	(0.18)	(0.19)	(0.18)	(0.17)	(0.17)
Green certification	−0.39 *	−0.19	0.22	0.21	0.90 ***
	(0.20)	(0.22)	(0.20)	(0.18)	(0.18)
Organic certification	0.31	0.47 **	0.27	0.32 *	0.48 ***
	(0.19)	(0.21)	(0.18)	(0.17)	(0.17)
Hazard-free certification	0.09	0.11	0.02	−0.07	−0.09
	(0.19)	(0.20)	(0.18)	(0.17)	(0.17)
Supermarket	−0.08	−0.24	0.09	0.07	0.10
	(0.18)	(0.19)	(0.17)	(0.16)	(0.16)
Consumer reads ingredients	−0.01	0.13	0.11	−0.06	−0.11
	(0.24)	(0.25)	(0.23)	(0.22)	(0.21)
Traceability	0.15	0.07	0.03	−0.16	−0.20
	(0.27)	(0.28)	(0.26)	(0.25)	(0.25)
Price safety cue	0.61 ***	0.56 ***	0.41 **	0.38 **	0.59 ***
	(0.17)	(0.18)	(0.16)	(0.15)	(0.16)
Constant	−1.02 ***	−1.33 ***	−1.24 ***	−0.70 ***	−0.68 ***
	(0.21)	(0.22)	(0.21)	(0.19)	(0.20)
Pseudo *R*^2^	0.03	0.03	0.02	0.02	0.07

Standard errors in parentheses; *** *p* < 0.01, ** *p* < 0.05, * *p* < 0.1.

**Table 4 foods-09-01153-t004:** Influence of certifications with age and education.

Variable	Coefficient				
	Fish	Meat	Milk	Eggs	Rice
SC logo	−0.42	−0.44	−0.62	−0.83 **	−0.80 **
	(0.41)	(0.42)	(0.41)	(0.40)	(0.41)
Green certification	−0.49	0.01	0.68	−0.05	0.74 *
	(0.46)	(0.48)	(0.45)	(0.43)	(0.43)
Organic certification	0.96 **	0.63	0.73	1.20 ***	0.93 **
	(0.46)	(0.48)	(0.46)	(0.45)	(0.45)
Hazard-free certification	−0.50	−0.26	−0.05	−0.13	−0.02
	(0.46)	(0.49)	(0.46)	(0.44)	(0.44)
Supermarket	0.16	−0.05	−0.25	0.43	0.55
	(0.40)	(0.42)	(0.40)	(0.38)	(0.38)
Consumer reads ingredients	−0.09	−0.05	0.78	−0.06	−0.69
	(0.57)	(0.58)	(0.55)	(0.54)	(0.55)
Traceability	0.34	−0.28	−0.51	−0.49	−0.32
	(0.69)	(0.78)	(0.72)	(0.67)	(0.65)
Price safety cue	1.32 ***	1.33 ***	1.07 ***	1.52 ***	1.82 ***
	(0.40)	(0.42)	(0.40)	(0.39)	(0.40)
Supermarket × age	−0.21	−0.06	−0.07	−0.78 **	−0.71 **
	(0.37)	(0.39)	(0.37)	(0.35)	(0.34)
Supermarket × education	−0.11	−0.12	0.58	0.04	−0.09
	(0.38)	(0.41)	(0.39)	(0.36)	(0.36)
Ingredient × age	0.10	0.03	−0.38	−0.02	−0.19
	(0.53)	(0.54)	(0.51)	(0.49)	(0.48)
Ingredient × education	0.03	0.23	−0.54	0.14	1.17 **
	(0.53)	(0.55)	(0.52)	(0.50)	(0.50)
SC × age	−0.00	−0.06	0.15	0.37	0.48
	(0.37)	(0.39)	(0.37)	(0.35)	(0.35)
SC × education	0.24	0.29	0.30	0.44	0.18
	(0.40)	(0.42)	(0.40)	(0.39)	(0.39)
Hazard-free × age	0.05	−0.00	−0.17	−0.62	−0.10
	(0.41)	(0.44)	(0.41)	(0.39)	(0.38)
Hazard free × education	0.82*	0.59	0.33	0.46	0.03
	(0.46)	(0.48)	(0.46)	(0.44)	(0.43)
Green × age	0.17	−0.04	0.03	0.83 **	0.73 *
	(0.43)	(0.47)	(0.43)	(0.40)	(0.40)
Green × education	0.01	−0.31	−0.74	−0.15	−0.18
	(0.46)	(0.49)	(0.45)	(0.43)	(0.43)
Organic × age	0.25	0.28	−0.39	−0.40	−0.77 **
	(0.44)	(0.46)	(0.43)	(0.40)	(0.39)
Organic × education	−1.01 **	−0.33	−0.36	−0.89 **	−0.18
	(0.47)	(0.49)	(0.46)	(0.45)	(0.44)
Traceability × age	−0.84	−0.66	−0.35	0.18	0.03
	(0.57)	(0.60)	(0.57)	(0.54)	(0.52)
Traceability × education	0.29	0.86	0.91	0.37	0.13
	(0.69)	(0.79)	(0.73)	(0.67)	(0.64)
Price safety cue × age	−0.33	−0.18	−0.41	−0.78 **	−1.02 ***
	(0.37)	(0.39)	(0.37)	(0.36)	(0.35)
Price safety cue × education	−0.86 **	−1.09 ***	−0.79 **	−1.17 ***	−1.11 ***
	(0.39)	(0.41)	(0.39)	(0.38)	(0.38)
Constant	−1.15 ***	−1.48 ***	−1.30 ***	−0.80 ***	−0.74 ***
	(0.22)	(0.24)	(0.22)	(0.21)	(0.20)
Pseudo *R*^2^	0.06	0.05	0.05	0.06	0.10

Standard errors in parentheses; *** *p* < 0.01, ** *p* < 0.05, * *p* < 0.1.

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
