# Peer review of "Urban Consumer Trust and Food Certifications in China"

_foods, 2020, doi:10.3390/foods9091153_

Round 1

Reviewer 1 Report

  1. Overall, I think the writing could be greatly improved throughout the manuscript. It is difficult to follow. E.g. line 46-49; paragraph starting on line 55 (reference 15 more clearly defines hazard free, green and organic food); line 64-66 (‘certification authority’ as opposed to ‘status’?); line 69 (do you mean knowledge about the concept of safe food?); line 84; line 140 etc.
  2. Please check the suitability of the references used throughout and add references where appropriate:

e.g. Line 32: reference [1]. It may be better to use the following reference which succinctly details scandals in the Chinese food industry as a whole: Qiao, G., Guo, T., & Klein, K. K. (2012). Melamine and other food safety and health scares in China: comparing households with and without young children. Food Control26(2), 378-386.

e.g. Line 33: Does reference 6 show that scandals have ‘limited the international development of the Chinese food industry’?

e.g. Line 35-40: perhaps this should be referenced? What does this mark actually indicate?

e.g. Line 43: perhaps the law documents should be referenced.

e.g. Line 55/27 etc: is it appropriate to use reference numbers in replace of author names?

e.g. Line 50: “(Ni, 2016)”?

  1. Line 69-78: Perhaps it would be more appropriate to incorporate this into section 2, starting on line 88.
  2. Line 79-83: The aim is unclear and ‘traceability’ is first introduced here.
  3. Line 88-130: While section two highlights selected research findings, it would be better to see a more succinct overview of the research. Collectively, what does the literature show? Please see section 6 of the following review document: Forum on Health, Environment and Development (FORHEAD), Working Group on Food Safety (2014). Food safety in China: a mapping of problems, governments and research. Also, these references may be useful: Hansstein, F. V. (2014). Consumer knowledge and attitudes towards food traceability: a comparison between the European Union, China and North America. In 2014 International conference on food security and nutrition IPCBEE, 67, 114-118; Liu, R., Pieniak, Z., & Verbeke, W. (2014). Food-related hazards in China: Consumers' perceptions of risk and trust in information sources. Food Control46, 291-298; Miller, S. A., Driver, T., Velasquez, N., & Saunders, C. M. (2014). Maximising Export Returns (MER): Consumer behaviour and trends for credence attributes in key markets and a review of how these may be communicated. AERU, research report no. 332.
  4. Section 3.1 (question design and survey): Was ethical approval obtained? I realize that the survey was detailed in the journal Food Control in February of this year. However, published details conflict with the design detailed within the current manuscript. Specifically, ‘Chinese urban areas’ (line 134) vs ‘The sample included 30% of rural households and 70% of urban households according to the proportion of urban and rural population in Zhejiang province’ (Food Control). Also, ‘the interviewers approached the respondents in several public areas’ (line 138) vs ‘The questionnaires are mainly obtained by household survey’(Food Control). More details on the questionnaire should be provided e.g. was it a written self-complete in Chinese? Were there inclusion/exclusion criteria? Overall, it would be good to see this section better structured (data collection and sample description; Questionnaire design and outline; measures specific to this analysis; data analysis with recoding) with details on the actual elicitation method. It is unclear what the participants were actually asked e.g. what does ‘know’ (line 157), ‘retain’ (line 158), ‘knows the certification of traceability’ (line 172), ‘supermarket age’ (table 3) etc. mean? Is the outcome ‘safety perception of the named food’?
  5. Results (starting on line 188). What are ‘panels’ (line 189)?
  6. Unfortunately, I am unable to comment any further on the results and discussion due to the issues highlighted in point 6.

Author Response

We would thank the anonymous reviewers for their comments and the appreciated suggestions for improving the quality of the manuscript.

We made all suggestions provided by reviewers; all changes are highlighted in red font in the text. Also, the responses to reviewers are typed in red in the .docx file

Reviewer 2 Report

The topic of the study is important and really interesting. The article provides deep insight and arrives at interesting results, however I have some remarks for the Authors.

 Introduction

The concept of the ‘green food’ and ‘organic food’ is not clear enough actually. Please describe this aspect in more specific way.

Moreover, explain the consumer’s perception of food quality. You have mentioned that the concept of ‘food safety’ and ‘food quality’ are correlated, however it should described in details.

Please rewrite also the aim of the paper in order to make it understandable/specific for the readers.

Referring to lines 50-51 (page 2) change the reference Ni, 2016 into the proper number of reference (pay attention to page 3 lines 91-92 also).

Chinese consumer’s knowledge and safety perception about food certification

However the WTP concept is known/popular in the scientific literature, please explain the acronym of ‘WTP’(page 3 line 110) and explain the main concept of ‘willingness to pay’ in order to catch the idea of ‘price safety cue’. It should be described more clear in your manuscript.

Results

You have motioned that ‘organic certification’ refers mainly to rice and eggs (page9, lines 227-228); please describe/explain also the aspect of fish - table 3, because it is not clear enough for the final reader.

Discussion and conclusions

The discussion section should include more references referring the whole concept. In fact, it should be more precise.

References

Check the reference no. 8, I think that something is missing here.

Author Response

(The authors gave the same response as above.)

Reviewer 3 Report

The most serious issue with the paper as presented, in my opinion, is the inadequate description and discussion of the interview schedule, the interviewer training process, sampling procedures/respondent selection, and data collection processes.

The authors refer readers to a previous publication based on the same data to get his information! At the very least this is inappropriate on the authors’ part.

The authors, in the data analysis, discussion and conclusions, seem to assume the “sample” data are representative of some unidentified population. Does the group of respondents represent a neighborhood of Chengdu City, Chengdu City as a while, Szechuan Province, China as a whole? Is the group of respondents a random sample, a stratified sample, another type of purposive sample, or is it collection of respondents selected on the basis of idiosyncratic and implicit criteria used by interviewers?

Another issue the authors need to address relates to the intended audience for this research paper. Is the intended primary audience academics or policy-makers? If it is academics, then the authors should make clear the theoretical, methodological, and or empirical contributions the study makes. The authors seem not to have explicitly addressed these issues.

If the intended audience is policy-makers then it needs to be made clear which policy makers are the intended audience. Is it local (city) officials, provincial officials, national officials? It seems that the maim audience appears to be unidentified officials and policy-makers and enforcers, but  they are not meaningfully identified.

Regardless of who the paper is intended for, the author need to address the issies of data and methods identified above.

The in-text referencing style is odd and inconsistent. This needs to be addressed.

The English is comprehensible, but not very good in general, and very “murky” in places. The paper could be substantially strengthened by English language copy-editing.  

Having said all this, I want to make it clear that I think the topic is both important and timely. The interconnected issues of food safety, security, and sovereignty are now inextricably entangled in the contradictory dynamics of national/local versus global/transnational food supply and value chains. Consumer understanding of, and confidence in, the sources and substance of food (and drug) quality and safety standards has never been more important, nor has it ever been more fraught uncertainty, distrust, and anxiety.

Social science researchers have potential to make significant contributions to our knowledge about consumer perceptions of the safety of their food, the adequacy of the safety standards, and the intelligibility, trustworthiness and utility of the food safety information provided to them. Addressing the issues I raised above will enable the authors to make a contribution to our knowledge in this area.

Author Response

(The authors gave the same response as above.)

Round 2

Reviewer 1 Report

Reviewer #1

(Reviewer responses in Italics)

1 Overall, I think the writing could be greatly improved throughout the manuscript. It is difficult to follow. E.g. line 46-49; paragraph starting on line 55 (reference 15 more clearly defines hazard free, green and organic food); line 64-66 (‘certification authority’ as opposed to ‘status’?); line 69 (do you mean knowledge about the concept of safe food?); line 84; line 140 etc.

Done. In order to ameliorate the manuscript and clarify better the difference between the certifications we added in the introduction table 1 that shows the comparison between hazard free food certification, green food certification and organic food certification. In addition, we clarify the concept of safe food for Chinese consumers.

Table 1 is a useful addition; however, I still feel that the ‘writing could be greatly improved throughout the manuscript’ and I am unsure about the meaning of several sentences. To give one example (now line 92): ‘consumer knowledge of safe food is still quite low’. Does ‘safe food’ mean hazard-free, green and organic certified food?

2 Please check the suitability of the references used throughout and add references where appropriate:

e.g. Line 32: reference [1]. It may be better to use the following reference which succinctly details scandals in the Chinese food industry as a whole: Qiao, G., Guo, T., & Klein, K. K. (2012). Melamine and other food safety and health scares in China: comparing households with and without young children. Food Control, 26(2), 378-386.

Done.

e.g. Line 33: Does reference 6 show that scandals have ‘limited the international development of the Chinese food industry’?

Done. We changed the word “development” with “reputation”. In detail the reference says “Since the last decade, a series of high-profile food safety scandals have tarnished the reputation of Chinese food products in international markets and have seriously shaken Chinese consumers' confidence in the domestic food industry.

This reference says the above and then references five other papers. Perhaps it would be more appropriate to reference original article(s) here.

e.g. Line 35-40: perhaps this should be referenced? What does this mark actually indicate?

Done. The QS system is short for Food Quality Safety Market Access system, which was implemented in a compulsive way by the government in mainland China. Since 2003, only after applying for this license, can food enterprises be allowed to enter the market for processing or trading (Bai et al., 2007). The Food and Drug Administration in 2015, stipulated that the QS system would be withdrawn from the food market within a three-year transition period and replaced by the food production marked with “SC” Permit Code (Liu, 2016). SC is the abbreviation of “Sheng Chan” (Sheng Chan means production). This change in policy emphasized the responsibilities of enterprises in ensuring food safety instead of government regulation.

e.g. Line 43: perhaps the law documents should be referenced.

Done.

Please check the reference list as they do not appear to be in the correct format.

e.g. Line 55/27 etc: is it appropriate to use reference numbers in replace of author names?

We insert automatically (software Mendeley) the reference considering the citation style of Foods. It is not possible to switch name/number in the same document.

Thank you for adding the author names in the manuscript.

e.g. Line 50: “(Ni, 2016)”?

Done.

3 Line 69-78: Perhaps it would be more appropriate to incorporate this into section 2, starting on line 88.

Done.

4 Line 79-83: The aim is unclear and ‘traceability’ is first introduced here.

Done. We rewrote the sentence in order to make it clear that other food-related attributes were also analyzed in the work.

I still think the aim is unclear, perhaps this is, in-part, due to the fact that ‘safe food products’ are not defined.

5 Line 88-130: While section two highlights selected research findings, it would be better to see a more succinct overview of the research. Collectively, what does the literature show? Please see section 6 of the following review document: Forum on Health, Environment and Development (FORHEAD), Working Group on Food Safety (2014). Food safety in China: a mapping of problems, governments and research. Also, these references may be useful: Hansstein, F. V. (2014). Consumer knowledge and attitudes towards food traceability: a comparison between the European Union, China and North America. In 2014 International conference on food security and nutrition IPCBEE, 67, 114-118; Liu, R., Pieniak, Z., & Verbeke, W. (2014). Food-related hazards in China: Consumers' perceptions of risk and trust in information sources. Food Control, 46, 291-298; Miller, S. A., Driver, T., Velasquez, N., & Saunders, C. M. (2014). Maximising Export Returns (MER): Consumer behaviour and trends for credence attributes in key markets and a review of how these may be communicated. AERU, research report no. 332.

Done. We insert in the paper the review document: Forum on Health, Environment and Development (FORHEAD), Working Group on Food Safety (2014). Food safety in China: a mapping of problems, governments and research, which is more linked to the topic of our paper.

While this reference has been inserted, I still think ‘it would be better to see a more succinct overview of the research. Collectively, what does the literature show?’

6 Section 3.1 (question design and survey): Was ethical approval obtained? I realize that the survey was detailed in the journal Food Control in February of this year. However, published details conflict with the design detailed within the current manuscript. Specifically, ‘Chinese urban areas’ (line 134) vs ‘The sample included 30% of rural households and 70% of urban households according to the proportion of urban and rural population in Zhejiang province’ (Food Control). Also, ‘the interviewers approached the respondents in several public areas’ (line 138) vs ‘The questionnaires are mainly obtained by household survey’(Food Control). More details on the questionnaire should be provided e.g. was it a written self-complete in Chinese? Were there inclusion/exclusion criteria? Overall, it would be good to see this section better structured (data collection and sample description; Questionnaire design and outline; measures specific to this analysis; data analysis with recoding) with details on the actual elicitation method. It is unclear what the participants were actually asked e.g. what does ‘know’ (line 157), ‘retain’ (line 158), ‘knows the certification of traceability’ (line 172), ‘supermarket age’ (table 3) etc. mean? Is the outcome ‘safety perception of the named food’?

We agree. We rewrote section 3.1 by including the requested missing information. In order to understand what the participants were actually asked, we upload the questionnaire as supplementary file.

This section is improved, however, I still think it would be appropriate to detail how the questions, relating to the variables in table 3, were asked.

7 Results (starting on line 188). What are ‘panels’ (line 189)?

By the word “Panel” we mean group of consumers (or retailers) on which surveys are made.

8 Unfortunately, I am unable to comment any further on the results and discussion due to the issues highlighted in point 6.

Additional comments:

  1. Line 74 reads ‘Notably, despite these certifications (hazard-free, green,and organic) being mainly related to the quality of food, they are strongly implicated with the safety...’. This is confusing because safety appears to be listed within the definition in Table 1.

Author Response

Thanks for your precious improvement. We highlighted in yellow your new suggestions and we replied to you by typing in red during the text and in the attached document.

Reviewer 2 Report

Dear Authors,

Thank you for your responses to my suggestions.

Kind regards

Author Response

Thanks for your efforts.

Reviewer 3 Report

None. My concerns as presented in the first round of reviews have been adequately addressed 

Author Response

Thanks for your efforts.